# Study on Low-Temperature Conductive Silver Pastes Containing Bi-Based Glass for MgTiO_3_ Electronic Power Devices

**DOI:** 10.3390/mi14091663

**Published:** 2023-08-25

**Authors:** Yunsheng Fu, Xianglei Yu, Li Liu, Xianjie Tang, Junpeng Li, Guoyou Gan

**Affiliations:** 1Faculty of Material Science and Engineering, Kunming University of Science and Technology, Kunming 650093, China; 20212230055@stu.kust.edu.cn (Y.F.);; 2Kunming Institute of Precious Metals State Key Laboratory of Advanced Technologies for Comprehensive Utilization of Platinum Metals, Kunming 650106, China; 3Sino-Platinum Metals Co., Ltd., Kunming 650106, China

**Keywords:** silver paste, MgTiO_3_, bonding strength, low-temperature sintering

## Abstract

Low-temperature lead-free silver pastes deserve thorough investigation for sustainable development and application of MgTiO_3_ ceramics in electronic devices. In this study, a series of Bi_2_O_3_-B_2_O_3_-ZnO-SiO_2_-Al_2_O_3_-CaO glasses with suitable softening temperatures were prepared via melt quenching using a type of micrometer silver powder formed by silver nanoparticle aggregates. The composite pastes containing silver powder, Bi_2_O_3_ glass powder and an organic vehicle were then screen-printed. The effects of glass powder concentration and sintering temperature on the microstructure of the surface interface were also investigated. The results showed that the silver paste for microwave dielectric ceramic filters (MgTiO_3_) possessed good electrical conductivity (2.28 mΩ/□) and high adhesion (43.46 N/mm^2^) after medium temperature (670 °C) sintering. Thus, this glass powder has great application potential in non-toxic lead-free silver pastes for metallization of MgTiO_3_ substrates.

## 1. Introduction

A microwave is an electromagnetic wave that is widely employed in radars, biological signal detection, and telecommunication systems due to its high transmittance, wide operating frequency, high directionality, large information capacity, and other advantages [1,2]. The rapid growth of the wireless communication technology market in recent years has fostered a significant demand for microwave ceramic materials that can be applied in oscillators, resonators, antennas and microwave substrates. For this reason, microwave dielectric ceramics have attracted a great deal of attention from researchers [3,4]. Among them, magnesium titanate ceramics (MgTiO_3_) with a rhombic symmetric titanite structure is a relatively cheap dielectric ceramic material that has been widely studied worldwide at the forefront of advanced schemes for its high Q-factor (16,000 GHz), outstanding stability, low loss, and small relative dielectric constant (εr = 17), and is a typical dielectric ceramic [5,6]. Numerous research studies have been undertaken on metal-to-ceramic connection technologies in order to employ ceramics in power electronics. At present, there are several types of ceramic metallization, such as molybdenum–manganese metallization [7], Direct Bonded Copper (DBC) and Active Metal Brazing (AMB) [8], as well as screen printing with thick film metallization [9]. Because of its excellent electrical conductivity and ability to sinter in air without the need for environment control, silver (Ag) has been widely employed as an electrode material for the ceramic surface [10].

Normally, ceramic dielectric stacks are sealed with a conductive paste made up of metal particles (Au, Ag, Cu), organic phases, and inorganic glass components. Choosing an organic vehicle that is compatible with the powder may offer the required thixotropy for screen printing. On the other hand, inorganic glass bonding agents that account for only a small portion of silver paste are also essential. Besides contributing to the sintering temperature of the paste, these additives may improve the conductivity of metal powder and strengthen the bonding between metal conductive paste and MgTiO_3_. Moreover, the composition, content, ratio and processing parameters of the conductive paste will determine the performance of the final electrode product. Nowadays, most of the glass powders available on the market which have the outstanding performance of silver pastes are leaded glass powders. However, due to detrimental effects on human health and the environment, many countries have banned the production of electronic components containing toxic substances, such as lead and mercury [11]. Therefore, soda lime glass as an eco-friendly glass powder has been used in some works, but its glass transition temperature is too high (>1000 °C) for large-scale fabrication. This is because a higher sintering temperature means an increase in industrial energy consumption [12]. Thanks to the similarity between Bi_2_O_3_ and PbO, the former can be a suitable substitute for PbO. There are many studies on Bi_2_O_3_-based glasses. For instance, Kim et al. [13]. explored the effect of Bi_2_O_3_ content on the sintering and crystallization behavior of lead-free low-temperature sintered Bi_2_O_3_-B_2_O_3_-SiO_2_ glasses. The results showed that both the glass transition temperature and glass crystallization temperature decreased to varying degrees as the Bi_2_O_3_ content increased. Kim et al. [14]. interpreted the behaviour of Bi_2_O_3_-ZnO-B_2_O_3_ nanoscale glass powders that underwent expansion at 370 °C as thermal decomposition of bismuth oxide and proposed that the glass transition, densification and glass softening temperatures of nanoscale glass powders were significantly lower than those of micron and submicron powders. Tsai et al. [12] investigated the influence of Bi_2_O_3_-based glasses on silver pastes and demonstrated that the thickness of the interfacial glass layer changed with the Bi_2_O_3_ glass content. The prepared glass allowed the uniform precipitation of silver grains, as well as the wetting of the silver surface and the silicon wafer. Bih et al. [15] prepared Bi_2_O_3_-BaO-P_2_O_5_ glass as a sintering aid and coating agent for BaTiO_3_ ceramics. It was established that the composites with 5% BBP glass addition had the highest dielectric constants and low porosity compared to pure ceramics. Guo et al. [16] successfully prepared and characterized Bi_2_O_3_-B_2_O_3_-ZnO glass which was afterward used to join two Al_2_O_3_ ceramics at 600–750 °C. The main product of the brazing was presented by ZnAl_2_O_4_ particles. Once the temperature rose, the penetration of the glass into the Al_2_O_3_ became more severe and the shear strength of the joint also varied. Numerous studies have demonstrated the potential of these glass systems for widespread applications, including solar cells, glass ceramics, and brazing connections. Meanwhile, complementary research is still needed on silver pastes that incorporate Bi_2_O_3_ glass in ceramic metallization so as to fully understand the related material properties under specific operation conditions.

Most manipulations with pastes require processing at 870 °C or even higher temperatures [17,18], and some of them also need special atmospheric [19] and pressure control tools [20], which undoubtedly increase the industrial cost and maintenance difficulty. Moreover, when exposed to higher sintering temperatures, the crystal structure of ceramics tends to undergo transformations, which may adversely affect the dielectric properties [21] and thermal shock resistance [22] of ceramic substrates. Since the electronics industry is moving toward energy saving, green technologies and cost-effectiveness, the ability to produce pastes with the desired characteristics for metallization of ceramic substrates deserves a thorough study. In this work, a micro powder composed of agglomerated silver nanoparticles was used as a conductive phase. In particular, a Bi_2_O_3_-B_2_O_3_-ZnO-SiO_2_-Al_2_O_3_-CaO glass with a suitable glass softening temperature (400–500 °C) and good adhesion to MgTiO_3_ substrate was prepared via melt quenching. The substrate was then metallized through a cheap and effective screen printing process. Silver paste with excellent electrical conductivity (2.28 mΩ/□), high adhesion (43.46 N/mm^2^) and medium sintering temperature (670 °C) was thus successfully obtained. The microstructures of silver films sintered on MgTiO_3_ substrates and the effects of sintering temperature and glass powder content on the electrical properties and adhesive strength of these films were also systematically investigated.

## 2. Experimental Methods

### 2.1. Materials

The raw materials used for the preparation of glasses were analytically pure reagents. including Bi_2_O_3_, B_2_O_3_, ZnO, SiO_2_, Al_2_O_3_, CaO powders (≥99 wt.%), and organic solvents (pine oil alcohol, dibutyl lead benzo-dicarboxylate, hydrogenated castor oil and Span 85), all purchased from Shanghai Aladdin Biochemical Technology Co., Ltd., Shanghai, China. MgTiO_3_ ceramics (5 mm × 20 mm × 20 mm) and ethyl-cellulose were supplied by Kunming SINO-PLATINUM METALS Co., Ltd., Kunming, China. Silver powder was purchased from Shanghai Forsman Technology Co., Ltd., Shanghai, China. The conductive phases containing silver powder with three different particle sizes and shapes were labeled as silver powder A, B, and C.

### 2.2. Synthesis of Glass Powder

Fractions with Bi contents of 30, 40, 50, 60 and 70 were referred to as A1, A2, A3, A4 and A5. The fractions of the samples are shown in Table 1.

Glasses were prepared via the traditional melt quenching method. First powders were ultrasonically treated for 0.5 h (using an LC-UC-100 system, YingJia Electronics Co., Ltd., YingJia, China) at a frequency of 40 ± 2 KHz and exposed to magnetic stirring (H97-A, Shanghai Hei-PLATE Co., Ltd., Shanghai, China) at a rotation speed of 300 rpm for 1.5 h. The powders were then dried at 80 °C for 3 h. The mixed and dried powders were afterward loaded into a 50 mL Al_2_O_3_ crucible. The molten material was then heated to 1300 °C in a muffle furnace (KSL1400, Hefei Kejing Materials Technology Co., Ltd., Hefei, China) at a rate of 10 °C/min and held at this temperature for 1.5 h to be transformed to a liquid. The final liquid glass was poured directly into deionized water to obtain clear yellow-brown glass particles. The dry glass particles were then combined with ethanol and subjected to wet ball milling for 10 h, 300 rpm. The powders were finally collected by screening the relevant glass powders through a 400 mesh screen.

### 2.3. Preparation of Silver Paste and Silver Film

First, an organic vehicle was prepared for dispersing and wetting the particles. The detailed composition of the organic vehicle is shown in Table 2. The corresponding components were weighed precisely in the required proportions, then mixed in a glass beaker and stirred well in a constant temperature oil bath using a magnetic stirrer at 60 °C until the ethyl cellulose was completely dissolved. The mixtures were finally stored at room temperature. To guarantee a lower contact resistance and to reduce the losses of the silver paste, the rheology and thixotropy of the paste needs to be modified to make the silver paste suitable for screen printing. The organic vehicle, glass powder (A5) and silver powder were afterward mixed in a mortar in the proportions shown in Figure 1a, and then finely dispersed through a three-roller machine to obtain a silver paste (Table 3). The paste was screen-printed on the MgTiO_3_ substrate and eventually sintered, conforming to the procedure illustrated in Figure 1b, to form a silver film. The heating from room temperature to 300 °C was implemented to eliminate the organic vehicle.

### 2.4. Materials Characterization

To study the phase composition of the glass, the X-ray diffraction analysis was performed at room temperature by means of an X-ray diffractometer (XRD, Ultima IV, Shanghai Bright Industrial Co., Ltd., Shanghai, China) using Cu Kα radiation in the 2θ range of 10–80°. The infrared (IR) transmission measurements were made over the range of 4000 to 400 cm^−1^ at room temperature using a FTIR spectrometer (Vertex 70, Bruker Beijing Baiduheng Instrumentation Co., Ltd., Beijing, China). The DSC curves were acquired on glass powders taken in a quantity of about 20 mg using a thermogravimetric analyzer (TA Discovery SDT 650 PerkinElmer, Waltham, MA, USA) under a nitrogen atmosphere in a temperature range of 25 to 800 °C at a heating rate of 5 °C/min. The square resistance of thick films was measured by a four-probe tester (ST-2258C, Suzhou Lattice Electronics, Suzhou, China). A scanning electron microscope (SEM, SU8010, Hitachi, Japan) equipped with an energy dispersive spectrometer (EDS) was employed to observe the microstructures of the surface and the silver film–substrate junction. The element distribution was visualized via energy dispersive spectroscopy (EDS). To conduct the push–pull tests, leads were soldered to a pad with a 2 mm diameter using a silver-plated copper wire (SnAg3.0Cu0.5), as shown in Figure 1c. The pad was then pulled out vertically by means of a push–pull tester (VICTOR, Shenzhen Yisheng Shengli Technology Co., Ltd., Shenzhen, China) under a load of 300 N. The average value was taken as the result of the test.

## 3. Results and Discussion

### 3.1. Characterization of the Glass Frits

The X-ray diffraction pattern of the glass powder is shown in Figure 2. The broad XRD peaks at 2θ of 20–40° and 40–60° confirmed the amorphous structure typical of borate glasses [23].

Figure 3 depicts the DSC curve of the glass in the range from 200 °C to 800 °C. The glass transition temperature *T_g_*, the initial crystallization temperature *T_c_* and the glass softening temperature *T_f_* of each sample are listed in Table 4. As can be seen from Table 4, there was a tendency for the glass transition temperature *T_g_* to decrease as the Bi_2_O_3_ content increased. The thermal stability Δ*T* of the glass can be expressed as the difference between the initial crystallization temperature and the glass transition point as follows [24]:△T=Tc−Tg

The larger the Δ*T* value, the greater the thermal stability of the glass.

The macroscopic properties of the glass are usually related to its microstructure which can be identified via infrared spectroscopy.

The infrared absorption spectrum of the glass sample is shown in Figure 4. All the constituents of the glass exhibited peaks at approximately 470 cm^−1^, 710 cm^−1^, 860 cm^−1^, 1000 cm^−1^, 1130 cm^−1^, and 1350 cm^−1^. Table 5 summarizes the possible vibration modes associated with glass structures. In the glass structure, the band range 430–520 cm^−1^ is a combination of Bi-O-Bi tensile vibration and Si-O bending vibration in the [BiO_6_] octahedral unit [23]. In the glass structure, bismuth ions exist in six and three coordination states. With the increase in Bi_2_O_3_ content, the strength gradually increased. In the borate glass, boron and oxygen ions exist in the form of a triangular coordination [BO_3_] and a tetrahedral coordination [BO_4_]. This is caused by bending vibrations of oxygen bridging between two tripartite boron atoms (BIII-O-BIII) in the range 690–730 cm^−1^ [25]. As soon as the B content increases, the absorption becomes significant in this range, being manifested by not only the tensile vibration of the Bi-O bond in the [BiO_3_] tetrahedron, but also the flexural stretching in the same tetrahedron unit. The range 830–900 cm^−1^ is attributed to the Bi-O symmetric stretching vibration in the [BiO_3_] tetrahedron. The gradual broadening of the spectral band at 830–900 cm^−1^ in the spectra of samples A1–A5 indicated that the amount of [BiO_3_] and [BiO_6_] units gradually increased with the increase in Bi_2_O_3_ content. Meanwhile, the decrease in the Bi_2_O_3_ amount was also very beneficial for strengthening the glass network structure. The spectral range of 990–1080 cm^−1^ is assigned to tensile vibrations of the B-O bonds from the tri-borate, tetraborate and penta-borate groups of [BO_4_] units [25,26]. With the increase in Bi_2_O_3_ content, the intensities of the peaks decreased and their centers moved toward the lower wave numbers. A peak at 1130 cm^−1^ is referred to as the B-O-B tensile vibration in the [BO_4_] unit [27]. The bands in the range 1300–1400 cm^−1^ are caused by the asymmetric stretching vibrations of the B-O bonds in the [BO_3_] unit [28]. With the increase in the Bi_2_O_3_ content, their intensities and widths decreased. Moreover, as the amount of B_2_O_3_ increased, the central band at 1380 cm^−1^ became wider and gradually shifted to the lower frequencies. Another peak observed at 1643 cm^−1^ can be attributed to the O-H bending vibration, which might be due to the moisture absorbed by glass powder in a humid environment [25,29]. In summary, the glass structure was mainly composed of [BiO_3_], [BiO_6_], [BO_3_] and [BO_4_] units. With the increase in Bi_2_O_3_ content, the conversion of [BO_3_] to [BO_4_] groups increased the amount of non-bridging oxygen species.

### 3.2. Effect of Silver Powder Category on Sintering Temperature

Figure 5 depicts a graph of the change in square resistance for different types of silver powders under the same sintering conditions. As can be seen from the figure, the square resistance of silver powder A was lowest (2.28 mΩ/□), while that of silver powders B and C after sintering increased to various degrees. Figure 6a–c show the surface morphologies of the three different types of silver powders. From Figure 6a, it can be seen that silver powder A was formed through the aggregation of nanoscale silver grains. According to Figure 6b, silver powder B was composed of spherical particles whose size (1.2–1.5 μm) was similar to those making powder A. As shown in Figure 6c, the silver powder C consisted of irregular sub-micron (0.3–0.5 μm) silver particles. Figure 6d–f display the surface morphologies of silver powders A, B and C under the same sintering conditions. Among them, silver powder A, with a special structure, exhibited a denser conductive network after sintering at medium temperature, while the other two powders had holes and laminations at lower sintering temperatures, which were not conducive to the formation of a dense conductive network. The results showed that powder A could ensure the more optimal sintering parameters at lower temperatures due to abundant silver nanoparticles on its surface. In particular, the high surface energy and low melting point allow the silver nanoparticles, to be easily clustered together by minimizing the surface energy [31], which aids in both linking the large particles and reducing the sintering temperature. Chiodi et al. [32] have reported that silver nanoparticles are more favorable for low-temperature attachment under oxygen-rich conditions. The present study, on the other hand, only needed to be carried out under air, which was conducive to the sintering of silver and was in line with the trend of low cost and energy saving, making the proposed method potentially utilizable.

### 3.3. Effect of Glass Powder Content on the Properties of Silver Film

The effect of ranged glass contents on the square resistance of the silver film at the corresponding sintering temperature is shown in Figure 7. It can be seen that, once the glass powder content increased from 1 wt.% to 3 wt.%, the square resistance decreased to its minimum value of 2.28 mΩ/□. Meanwhile, as the glass powder content continued to increase, the square resistance tended to increase linearly. At the glass powder content of 11 wt.%, the square resistance decreased slightly. The square resistance will vary depending on the microstructure of the silver coating. Figure 8 displays the sintered surfaces of silver films with various glass contents at the same sintering temperature. A comparison of the microstructures in Figure 8 revealed that, as the glass content increased, the silver grains grown by diffusion and adhesion formed a dense film. As shown in Figure 8a, at the glass content of 1 wt.%, the silver particles were slightly molten and the linking between them became relatively weak, revealing gaps. It is noteworthy that the sintering enhanced the growth of particles, but did not lead to their shrinkage. Therefore, the silver particles in Figure 8a are only pressed together by the roughening after sintering without forming an effective connection, which accounts for their large square resistance. As soon as the glass content increased to 3 wt.% as shown in Figure 8b, the silver particles grew significantly, reigniting the driving force for densification, whereas pores and imperfections were significantly reduced [33]. The reinitialization of the driving force would have stimulated the grain growth and the formation of a more compact and dense structure of the silver film. This caused the silver particles to join together during sintering with no gaps between them, thereby forming a dense conductive network. As a result, the square resistance of the silver film with a glass content of 3 wt.% achieved a minimal value of 2.28 mΩ/□. With a further increase in glass content, the square resistance of the film continued to grow. As shown in Figure 8c,d, the excess glass acted as an insulator, preventing the growth and connection between the silver particles, which led to an increase in the square resistance of the film [34]. The densification of the sintered film was mainly affected by the flow of the liquid glass phase and the rearrangement of the particles. In a word, a too high or too low amount of glass did not allow production of a dense conductive silver film, promoting the formation of a porous structure, which resulted in many defects in the product. Solderability is also an essential indicator for evaluating the quality of silver films. For instance, silver films with glass powder contents of 1 wt.%, 3 wt.%, and 5 wt.% possessed excellent solderability. However, with excessive glass content, the glass phase on the surface of the silver layer will strongly suppress the reaction between the silver and the solder [35], The glassy phase present in the silver grain boundaries may also obstruct or ease the penetration of the solder into the electrode bulk. The higher the glass content, the harsher the soldering properties of the silver layer. In that regard, silver layers with glass content of 7 wt.%, 9 wt.%, and 11 wt.% exhibited good solderability. In summary, the conductive silver film with a glass concentration of 3 wt.% possessed the best overall performance, including low square resistance and reliable solderability.

### 3.4. Effect of Sintering Temperature on the Properties of Silver Film

Figure 9a illustrates the effect of sintering temperature on the thin layer resistance of a silver film with a glass powder content of 3 wt.%. It can be seen that the thin layer resistance had a tendency to drop as the sintering temperature increased, reaching a minimum value of 2.35 mΩ/□ at 670 °C. Figure 9b depicts the bond strength of the silver film containing 3 wt.% glass powder at various sintering temperatures. The bond strength of the film on the substrate firstly increased with the increase in sintering temperature, achieving a maximum of 43.46 N/mm^2^ at 670 °C, and then declined with a further increase of temperature.

The evolutionary relationship between the square resistance, adhesion and temperature of the silver film is defined by the microstructure morphology of the silver film after sintering. Figure 10 illustrates the SEM images of the surface morphologies and the cross-sections of the silver films obtained by 20-min sintering of the silver paste with 3 wt.% glass powder at 570 °C, 620 °C, 670 °C, and 720 °C. It is evident from Figure 10a–d that a large number of holes existed on the surface of the film sintered at 570 °C. This is because the sintering temperature was too low to allow the liquid phase of the glass powder to fully penetrate the entire network of silver particles and mix with them, yielding an inability to form a high-quality silver conductive network. Once the sintering temperature was increased to 620 °C, the surface pores were reduced to a large extent and the size of silver particles increased significantly, but a few pores still existed. It was suggested that, as the temperature increased, the liquid phase of the glass was further refined and the silver particles grew at higher temperatures which, however, were not yet optimal for sintering. As soon as the sintering temperature rose to 670 °C, a uniform and completely densified conductive network formed on the film surface. This indicates that, at sufficiently high temperatures, the glass phase can fully wet the silver powder, thus promoting the redistribution of the silver powder and forming a more dense film [29]. Meanwhile, when the sintering temperature reaches 720 °C, the surface is no longer dense and the glass phase is unevenly distributed within the silver network, forming many microporous structures and reducing the electrical conductivity of the whole film. It can be seen from the figure that the silver layer became denser as the sintering temperature increased, and some silver clumps were formed by coarsening of the silver grains at higher sintering temperatures.

The bonding interface between the silver layer and the MgTiO_3_ substrate after heat treatment at different temperatures can be clearly observed in Figure 10e–h. At the sintering temperatures of 570 °C and 620 °C, there was obvious delamination at the interface and little porosity in the silver layer. Among them, the lower sintering temperature prevented the glass material from completely softening and left it in the form of rigid inclusions which further hindered the overall shrinkage and reduced the adhesive strength of the silver film [36]. Once the sintering temperature reached 670 °C, a denser interface without any cracks was observed. Moreover, the silver film and the substrate were well bonded, indicating that the glass material was completely liquefied at 670 °C and possessed excellent wettability to the MgTiO_3_ substrate. In addition, the silver layer became denser and uniform, allowing the silver film to bind tightly to the MgTiO_3_ substrate. Thus, the compact silver layer improved the bond strength of the joint [37,38,39,40]. However, when the sintering temperature was raised to 720 °C, cracks reappeared at the interface, and numerous pores also emerged inside the silver layer, which degraded the bonding between the silver film and the substrate. In this respect, EDS analysis was performed to inspect the distribution of the relevant elements at the interface near the silver layer and the MgTiO_3_ substrate after sintering at 570 °C (Figure 11a) and 670 °C (Figure 11b). Based on the results, it was concluded that Bi Si, Ca, elements as part of the glass phase were uniformly distributed in the silver layer after heat treatment at the temperature of 570 °C. Once the heat treatment temperature rose to 670 °C, the glass elements flowed into the MgTiO_3_ substrate. It was thus demonstrated that the glass powder could form a well-connected interface between the ceramic substrate and the silver layer at the temperature of 670 °C.

## 4. Conclusions

In order to realize the metallization of MgTiO_3_ substrates at lower temperatures, a micro-powder formed by silver nanoparticle aggregates was selected as the conductive phase in this work. Silver paste containing lead-free glass powder was produced by firing silver paste with excellent electrical conductivity (2.28 mΩ/□) and high adhesion (43.46 N/mm^2^) on a MgTiO_3_ substrate at a medium temperature (670 °C) conventional melt quenching of Bi_2_O_3_-B_2_O_3_-ZnO-SiO_2_-Al_2_O_3_-CaO glass prepared at a suitable softening temperature. The surface morphologies of the silver film and the interfacial layer between the silver film and the MgTiO_3_ substrate were analyzed via scanning electron microscopy and energy dispersive spectroscopy. It was found that the silver film had a dense structure and was well-bonded to the MgTiO_3_ substrate. This meant that the prepared glass possessed excellent fluidity and wettability during the firing process. The satisfactory solderability and reliable adhesion of related silver layers MgTiO_3_ substrates, as well as good electrical conductivity at lower sintering temperatures, are the most significant properties for the ceramic surface metallization industry. The resulting glass powders can be considered promising for the encapsulation of high-performance power electronic devices.

## Figures and Tables

**Figure 1 micromachines-14-01663-f001:**
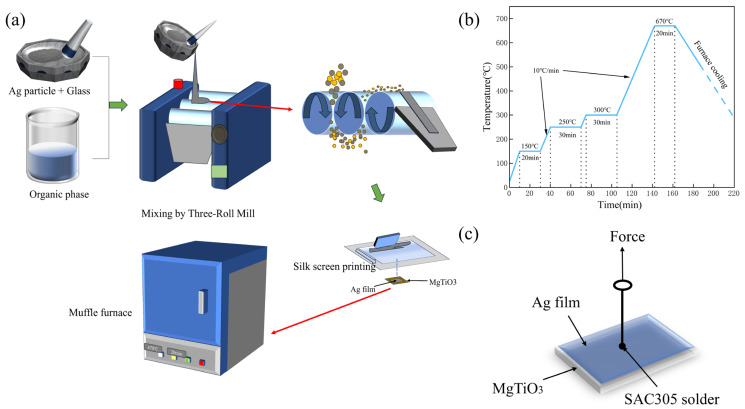
(**a**) The preparation process of silver paste; (**b**) silver film sintering curve; (**c**) drawing test diagram for adhesion test.

**Figure 2 micromachines-14-01663-f002:**
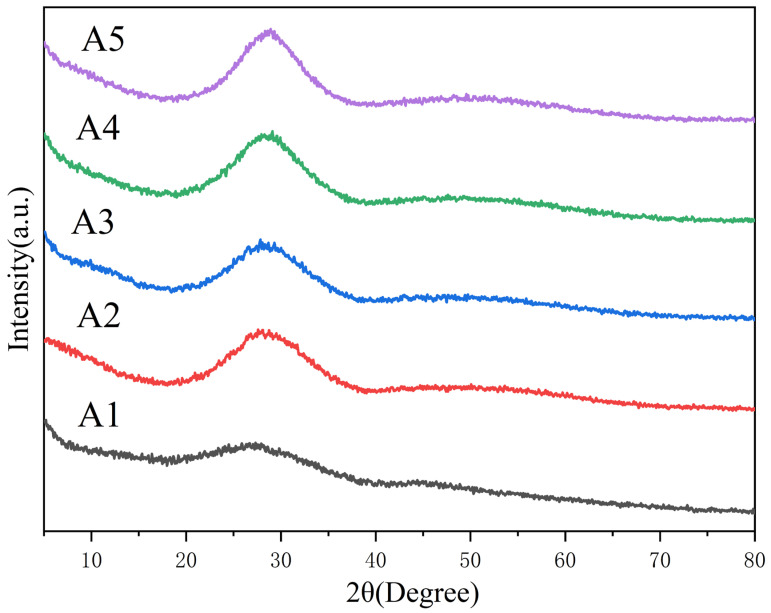
XRD patterns of glass samples.

**Figure 3 micromachines-14-01663-f003:**
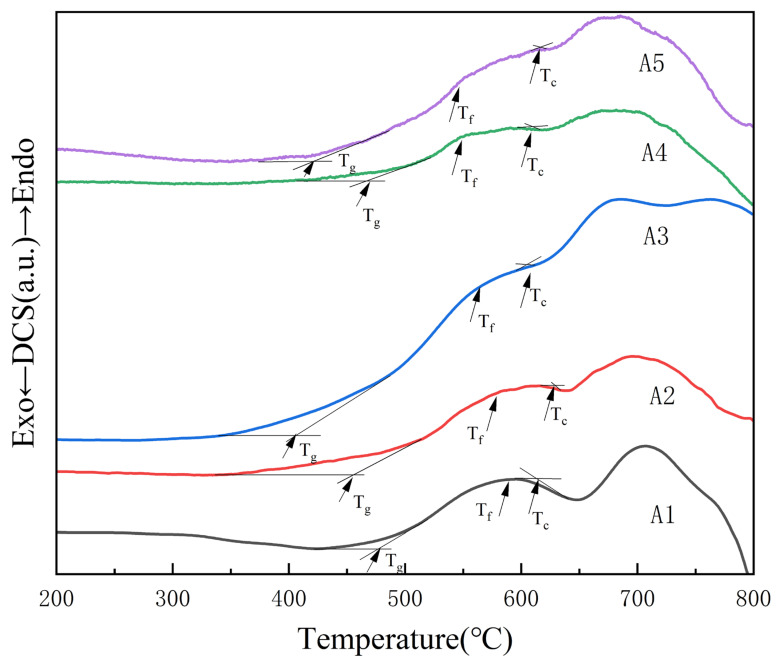
DSC curves of all the glass samples.

**Figure 4 micromachines-14-01663-f004:**
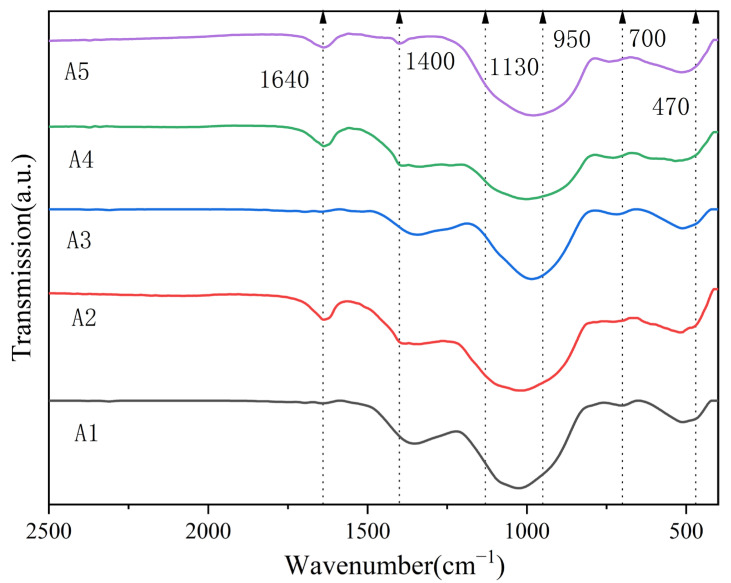
Infrared absorption spectra of glass samples.

**Figure 5 micromachines-14-01663-f005:**
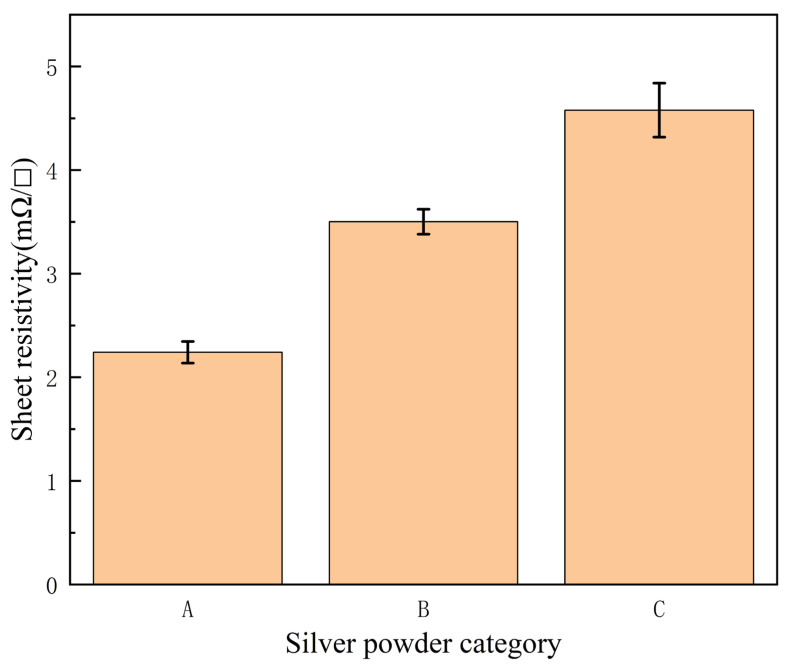
Comparison chart of different silver powder sintered sheet resistances.

**Figure 6 micromachines-14-01663-f006:**
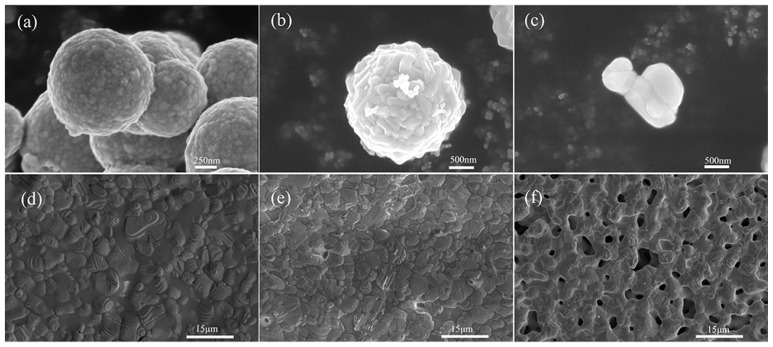
The shape of different silver powders: (**a**) Silver powder A; (**b**) Silver powder B; (**c**) Silver powder C and Sintered surface diagram of different silver powders: (**d**) Silver powder A; (**e**) Silver powder B; (**f**) Silver powder C.

**Figure 7 micromachines-14-01663-f007:**
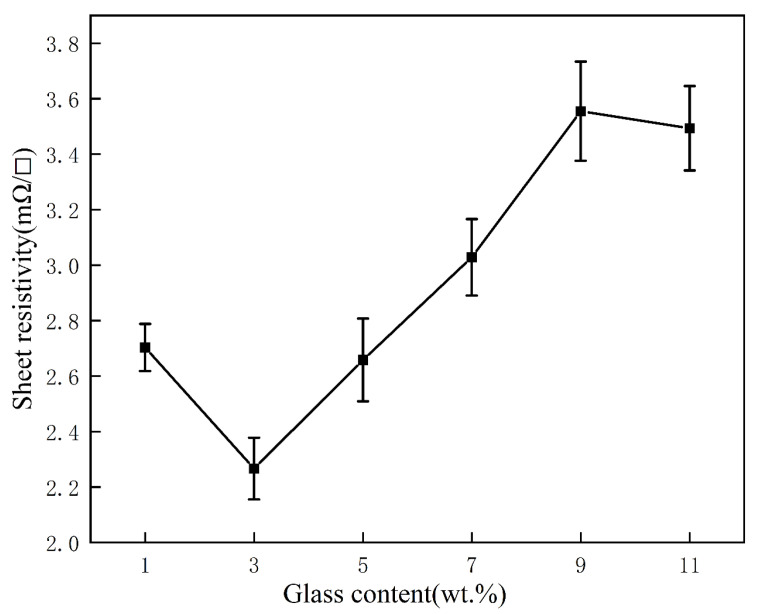
Sheet resistivity of the Ag films with different glass contents.

**Figure 8 micromachines-14-01663-f008:**
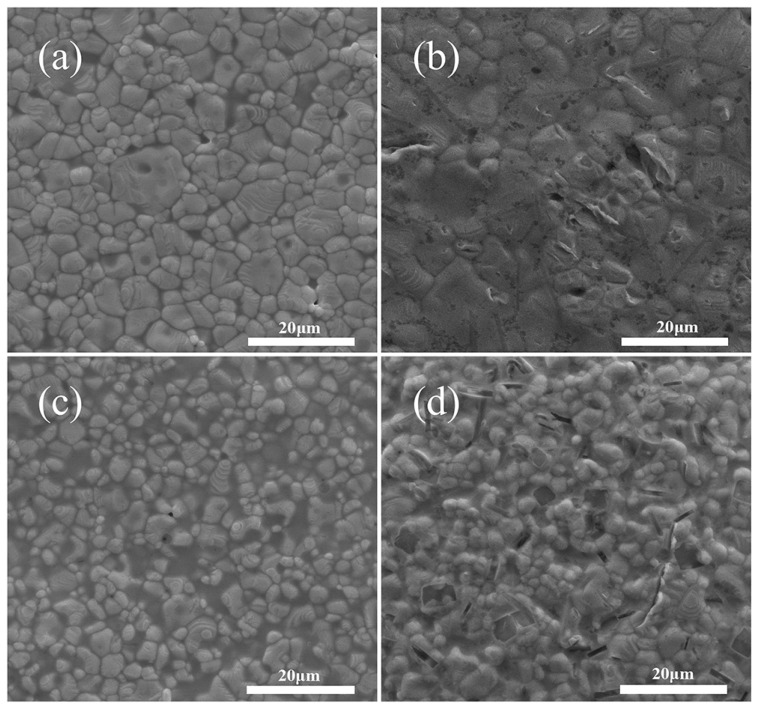
Surface SEM images of the Ag films containing (**a**) 1 wt.% (**b**) 3 wt.% (**c**) 5 wt.% (**d**) 7 wt.% glass.

**Figure 9 micromachines-14-01663-f009:**
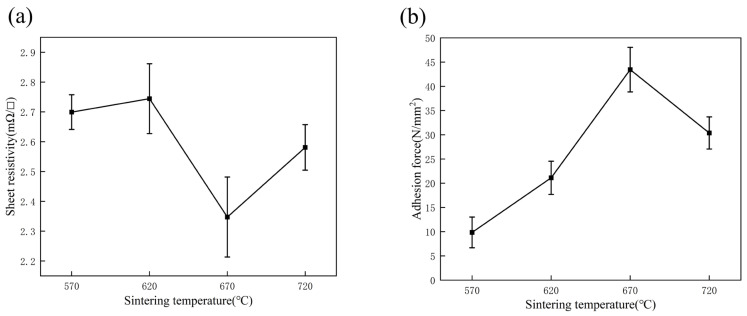
(**a**) Sheet resistivity of the Ag films containing 3 wt.% glass. (**b**) Adhesion force of the Ag films containing 3 wt.% glass and sintered at 570 °C, 620 °C, 670 °C, 720 °C.

**Figure 10 micromachines-14-01663-f010:**
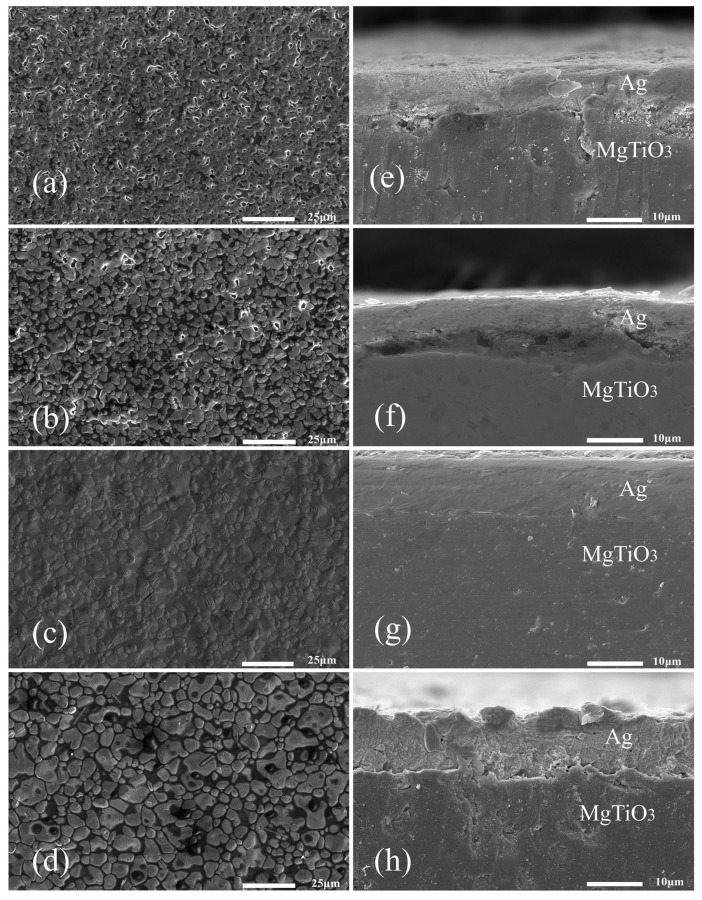
SEM images of silver films sintered at (**a**) 570 °C, (**b**) 620 °C, (**c**) 670 °C, (**d**) 720 °C. Cross-section SEM photographs of silver film at different temperatures (**e**) 570 °C, (**f**) 620 °C, (**g**) 670 °C, (**h**) 720 °C.

**Figure 11 micromachines-14-01663-f011:**
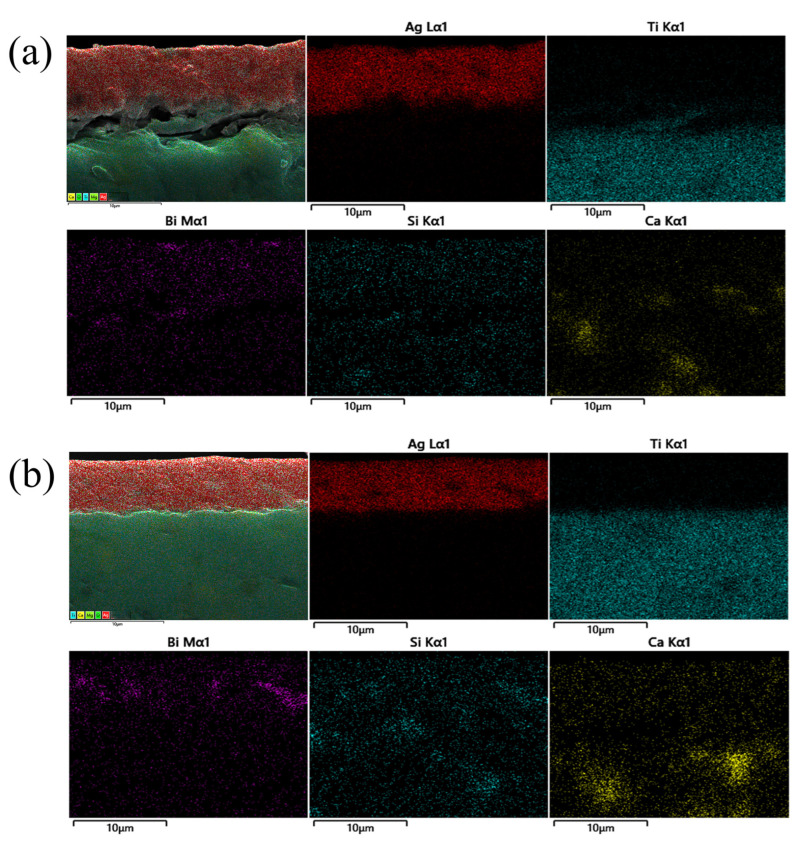
EDS analysis was performed on the distribution of elements in the cross section of the silver layer containing 3% glass powder on MgTiO_3_ substrate after sintering at (**a**) 570 °C and (**b**) 670 °C for 20 min.

**Table 1 micromachines-14-01663-t001:** Compositions of the glass system (wt.%).

Sample	Bi_2_O_3_	B_2_O_3_	SiO_2_	ZnO	Al_2_O_3_	CaO
A1	30	40	15	9	3	3
A2	40	30	15	9	3	3
A3	50	20	15	9	3	3
A4	60	10	15	9	3	3
A5	70	0	15	9	3	3

**Table 2 micromachines-14-01663-t002:** Compositions and content of the organic vehicle (wt.%).

Component	Ethyl Cellulose	Terpineol	Dibutyl Phthalate	Hydrogenated Castor Oil	Siban 85
Content	12	56	30	0.5	1.5

**Table 3 micromachines-14-01663-t003:** Compositions of the Ag paste (wt.%).

Sample	Ag	Glass Frit	Organic Vehicle
#1	82	1	17
#2	80	3	17
#3	78	5	17
#4	76	7	17
#5	74	9	17
#6	72	11	17

**Table 4 micromachines-14-01663-t004:** Characteristic temperatures of the glass.

Sample	*T_g_* (°C)	*T_f_* (°C)	*T_c_* (°C)	Δ*T* = *T_c_* − *T_g_* (°C)
A1	477	587	614	137
A2	452	583	627	175
A3	408	565	603	195
A4	465	547	609	144
A5	421	544	615	194

**Table 5 micromachines-14-01663-t005:** FT-IR peak assignments for the glass samples.

Band Position (cm^−1^)	Assignment
430–520	Bi-O-Bi in the [BiO_6_] octahedral unit [23]
690–730	Bending vibrations of oxygen bridges between two [BO_3_] units (BIII-O-BIII) [25]
830–900	Symmetric stretching vibrations of Bi-O bonds in [BiO_3_] pyramidal units [30]
990–1080	Stretching vibrations of B-O bonds in [BO_4_] units from tri-, tetra-, and penta-borate groups [25,26]
1130	B-O-B stretching vibration in [BO_4_] [27]
1300–1400	Asymmetric stretching of B-O bond in [BO_3_] units [28]

## Data Availability

Not applicable.

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
