# Peer review of "Study on Low-Temperature Conductive Silver Pastes Containing Bi-Based Glass for MgTiO3 Electronic Power Devices"

_micromachines, 2023, doi:10.3390/mi14091663_

Round 1

Reviewer 1 Report

Authors prepared a thick film silver paste for MgTiO3 substrate application. This work focused on the influence of Bi2O3 glass and sintering temperature. However, the manuscript is not well organized. The following points should be properly addressed before consideration for publication.

1. The title is too general. The introduction extensively mentions the necessity of studying Bi2O3 glass for silver paste, as well as sintering temperature. They are the main points of this article, but not reflected in the title.

2. The expression is not rigorous in many places, such as, “organic vehicle”, not “organic carrier”, nor “carrier organics”; “paste”, not “slurry”; and so on.

3. Punctuation symbols are used incorrectly in many place, such as, “at 300 r/min for 1.5h, The powder is then dried at 80 ℃ for 3h.” in line 15 page 3 should be “at 300 r/min for 1.5h. The powder is then dried at 80 ℃ for 3h.”. Capital and case are not distinguished, for example, “the intensity decreases and the bandwidth Narrows.” in line 12 page 7 should be “the intensity decreases and the bandwidth narrows.”. Superscript and subscript are used incorrectly in Table 5 page 8. And full spelling and abbreviation are mixed, such as, “Figure 8a” in line 9 page 10 and “Fig. 8a” in line 13 page 10.

4. The last sentence of section 2.3, page 3, “In this case, 20 minutes at 150 °C is used to eliminate the organic carrier.”, is not correct. Organic vehicles cannot be removed at 150 ℃. Please refer to the thermal decomposition temperature of ethyl cellulose.

5. In section 3.2, there is no significant difference between Figures 6a and b, making it difficult to conclude that powder A is coated with a layer of silver nanoparticles, while powder B does not. Please provide further evidence.

6. The description of Figure 7 in the second and third sentences on page 10 is not accurate.

7. On page 10, authors described that “The bonding strength of the solder pulled off from the silver layer by pulling test is greater than 10.8 N/mm2, which suggests that the silver layer has dependable solderability and good bonding with the substrate.”. Why 10.8 N/mm2? Where do the standards come from? Alternatively, please provide references.

The manuscript should be improved with the help of native English speaker.

Author Response

请参阅附件。

Reviewer 2 Report

The submitted manuscript is entitled Study on the preparation of thick film silver paste for MgTiO3 electronic power devices and its performance at the interface.

I found this study not carefully prepared and the manuscript needs revision for further consideration.

The authors do not completely pay attention to capital letters, indexes, and punctuation, which makes the readability of the manuscript very difficult.

The abstract should be more concise, i.e., I recommend rewriting the first sentence to improve readability.

The unit of electrical conductivity is wrong.

The unit of adhesion is wrong.

Units should be unified, for example, r/min, rpm, etc.

More details on the experiments should be provided.

What was the temperature of the XRD analysis?

The main concern is the DSC measurement. The authors should provide more details on the experiment, including the mass of samples.

It seems that the determination of the discussed temperatures requires processing of the presented curves.

How were the temperatures of glass transition and crystallization determined?

What do the authors mean by glass softening temperature Tf? Why is this temperature denoted as Tf?

Section 3.1

The X-ray diffraction pattern of the glass powder is shown in Figure 2. The broad peaks in the spectrum confirm the amorphous nature of the glass.

The glass is an amorphous solid. Why would the authors confirm this?

The coarsening of the particles during the sintering process enhances the growth between the particles, but does not lead to their shrinkage.

What does that growth mean? What is the mechanism of this process?

The silver particles are joined together by sintered growth and there are no more gaps between the silver particles, forming a dense conductive network.

What kind of networks are formed?

The pores act as defects in the film and tend to cause stress concentration and crack extension, leading to low stress brittle fracture.

The porosity should be determined to discuss this effect.

Round 2

Reviewer 1 Report

The manuscript was improved and is acceptable for publication.

Reviewer 2 Report

The authors changed the title of the revised manuscript. However, the present title is not informative. 

There are a number of problems with indexes and units.

The thermal stability ΔT of the glass can be expressed as the difference between the initial crystallization temperature and the glass transition point as follows: 

△?=??−??

How was the crystallization temperature (Tc) determined?

Is this the onset or peak temperature?  

Tc in Figure 3 is neither the onset nor the peak temperature.

Round 3

Reviewer 2 Report

The authors revised the manuscript appropriately. During the formatting process, you should pay attention to the units, in some cases these appear with the ‘□’ symbol.